# The R2R3-MYB Transcriptional Repressor *TgMYB4* Negatively Regulates Anthocyanin Biosynthesis in Tulips (*Tulipa gesneriana* L.)

**DOI:** 10.3390/ijms25010563

**Published:** 2024-01-01

**Authors:** Xianmei Hu, Zehui Liang, Tianxiao Sun, Ling Huang, Yanping Wang, Zhulong Chan, Lin Xiang

**Affiliations:** National Key Laboratory for Germplasm Innovation & Utilization of Horticultural Crops, College of Horticulture and Forestry Sciences, Huazhong Agricultural University, Wuhan 430070, China; xiaofeifei@webmail.hzau.edu.cn (X.H.);

**Keywords:** anthocyanin, R2R3-MYB, repressor, tulip, bHLH

## Abstract

Anthocyanins play a paramount role in color variation and significantly contribute to the economic value of ornamental plants. The conserved activation complex MYB-bHLH-WD40 (MBW; MYB: v-myb avian myeloblastosis viral oncogene homolog; bHLH: basic helix–loop–helix protein; WD40:WD-repeat protein) involved in anthocyanin biosynthesis has been thoroughly researched, but there have been limited investigations into the function of repressor factors. In this study, we characterized *TgMYB4*, an R2R3-MYB transcriptional repressor which is highly expressed during petal coloration in red petal cultivars. *TgMYB4*-overexpressing tobaccos exhibited white or light pink petals with less anthocyanin accumulation compared to control plants. TgMYB4 was found to inhibit the transcription of *ANTHOCYANIDIN SYNTHASE* (*TfANS1*) and *DIHYDRO-FLAVONOL-4-REDUCTASE* (*AtDFR*), although it did not bind to their promoters. Moreover, the TgMYB4 protein was able to compete with the MYB activator to bind to the :bHLHprotein, thereby suppressing the function of the activator MBW complex. These findings demonstrate that *TgMYB4* plays a suppressive role in the regulation of anthocyanin synthesis during flower pigmentation.

## 1. Introduction

Anthocyanins are flavonoid compounds that are commonly present in various parts of plants, such as the flowers, fruits, leaves, roots, and stems, adding to their vivid hue [1]. These vibrant colors are crucial for attracting pollinators, facilitating seed dispersion, and providing protection against damage from ultraviolet radiation [2,3,4]. Anthocyanin pigments have been the subject of extensive research, leading to notable scientific advancements over the last 150 years [5]. 

The anthocyanin biosynthesis pathway has been extensively characterized, and the structural genes responsible for encoding the enzymes in the biochemical reactions for anthocyanin synthesis have been identified and isolated from different plant species [6]. Transcription factors play a crucial role in regulating the transcriptional expression of these structural genes. Specifically, the highly conservative MBW complex, composed of R2R3-MYB transcription factors, bHLH transcription factors, and WD40 proteins, serves as the core of the anthocyanin regulatory network [5,7,8]. MBW is a highly organized complex, wherein each subunit possesses specific functions. MYB proteins directly bind to the promoters of structural genes to regulate gene transcription; bHLH proteins typically interact with MYB proteins to enhance regulation; and WD40 proteins primarily contribute to the stability of the complex [9,10,11,12].

In recent years, considerable numbers of anthocyanin repressors have been identified in plants, including various proteins and small RNA families [8]. MYB proteins are particularly common in flowering plants, with a wide array of anthocyanin activators and inhibitors. MYB repressors mainly include subgroup 4 (S4) R2R3-MYBs, S4-derived R3-MYB, and CPC-type R3-MYB [8,13,14,15,16,17]. Repressor S4 R2R3-MYB proteins share the R3 domain (‘(D/E) Lx2 (R/K) x3Lx6Lx3R’) with anthocyanin-activating subgroup 6 R2R3-MYBs at the N-terminal, which facilitates their interaction with bHLH proteins [9,13,18,19,20]. A conserved ERF motif (‘pdLNL(D/E) L’ or ‘DLNxxP’) at the C-terminal is considered as the repressor motif in plant R2R3-MYB proteins [21,22,23,24]. The S4-derived R3-MYB contains a fully functional R3 MYB domain and a partially truncated R2 MYB domain at the N-terminus. It possesses a degenerate EAR motif and a distinctive repression motif ‘TLLLFR’, located towards the end of its C-terminal [18,21,25,26]. CPC-type R3-MYB repressors are relatively small, consisting of 80–100 amino acids, and contain no repressive motifs but a conserved bHLH-interacting motif. They play a role by competing with the bHLH protein in the anthocyanin-activating MYB-bHLH-WD40 complex [9,27].

The highly conserved bHLH-binding motif and ERF motif are identified as common characteristics in MYB repressor proteins, which may determine the suppression function. The EAR motif is identified as the most predominant form of the transcriptional repression motif in plants [22]. In numerous studies, mutation or deletion of the EAR motif has demonstrated an essential role in the repressive activity of subgroup 4 R2R3-MYBs [13,19,28,29]. Meanwhile, the bHLH-binding motif is particularly important for MYB repressors to compete with MYB activators. PrtMYB182 in poplar, lacking a functional bHLH-binding motif, was unable to exert repression, while mutations of the ERF domain did not have any impact on its repressive activity [25]. In contrast, the repressive function of MdMYB16 in apples is dependent on the presence of the ERF domain rather than the bHLH-interacting motif [13]. The significance of the EAR motif and protein–protein interactions in the repressive activity of subgroup 4 R2R3-MYBs has been documented, and there are notable variations across different species [9,13,19,20,28,29,30,31].

Based on phylogenetic relationships, R2R3 MYBs repressors are classified as two types: ‘*FaMYB1*-like’ and ‘*AtMYB4*-like’ [24,32]. ‘*FaMYB1*-like’ repressors typically suppress the expression of key genes in the flavonoid pathway, such as *chalcone synthase* (*CHS*), *chalcone isomerase* (*CHI*), *flavanone-3-hydroxylase* (*F3H*), *dihydro-flavonol-4-reductase* (*DFR*), and *anthocyanidin synthase* (*ANS*), to limit the production of anthocyanins and PAs [25,29,33,34]. On the other hand, ‘*AtMYB4*-like’ MYB genes primarily regulate general phenylpropanoid steps, including phenylalanine ammonia-lyase (PAL), cinnamate 4-hydroxylase (C4H), 4-coumarate-CoA ligase (4CL), and other enzymes, to reduce the accumulation of phenolic acids and lignans [28,35,36,37]. In the last two decades, anthocyanin-repressing R2R3-MYB proteins have been identified in many species. *FaMYB1* was the first transcription repressor of anthocyanin synthesis isolated from strawberries [31]. *PhMYB27* is an anthocyanin repressor in petunias with a high expression under non-inductive shade conditions [38]. In *Vitis vinifera*, *VvMYB4a* and *VvMYB4b* can repress the transcription of genes in phenylpropanoid biosynthesis pathway and reduce the amount of low-molecular-weight phenolic compounds in transgenic petunia plants, whereas *VvMYBC2-L1* and *VvMYBC2-L3* can reduce the accumulation of anthocyanins in petals and proanthocyanidins in seeds [18]. *PtrMYB182* was discovered to inhibit the expression of structural and regulatory flavonoid genes, thereby exerting negative regulatory effects on flavonoid synthesis [25].

Anthocyanin constitutes the principal component of floral pigmentation, which plays a crucial role in the ornamental value of flowers. The regulatory mechanisms of anthocyanin biosynthesis though the MYB-bHLH-WD40 complex have been well indicated in many ornamental plants. Additionally, several anthocyanin repressors have been isolated in recent years, including *PhMYB27* and *PhMYBX* in petunias [9], *CmMYB21* in chrysanthemums [39], *LhR3MYB1/2* in lilies [14], *MlROI1* and *MlRTO* in *Mimulus* [15,40], *FhMYBX* in *Freesia hybrida* [41], and *MaMYBx* in *Grape hyacinths* [16]. In this study, TgMYB4 was identified as a subgroup 4 R2R3-MYB, which contains a bHLH-binding motif at the N-terminal and an ERF domain at the C-terminal. A functional assay revealed that it negatively regulates anthocyanin synthesis by inhibiting the transcription of structural genes. Additionally, the protein interaction between TgMYB4 and TgbHLH42-1 was characterized, which plays a crucial role in its repressive activity.

## 2. Results

### 2.1. Identification of TgMYB4: A Repressor Transcription Factor in Tulip Petals

An R2R3-MYB transcription factor was isolated from tulip petals, which contained a 627 bp open reading frame (ORF) and encoded 208 amino acids. In a phylogenetic analysis, it showed the closest evolutionary relationship with AmMYB308 in *Antirrhinum majus* and was classified into the *AtMYB4*-like group, and was thereby named TgMYB4 (Figure 1A). Amino acid sequence analysis of TgMYB4 revealed that the N-terminal R2R3 domain was highly conserved and included a bHLH-binding domain (Figure 1A). Two conserved motifs, namely the C1 motif (LlsrGIDPxT/SHRxI/L) and C2 motif (pdLNLD/ELxiG/S), were located in the C-terminal region. In addition, the C2 motif also serves as a diagnostic marker for subgroup 4 MYB proteins and contains the EAR-repressor domain (Figure 1B).

The TgMYB4-YFP and nuclear marker (VirD2NLS-mCherry) were co-transformed into *N. benthamiana* leaves for fluorescence observations. The YFP signal was primarily localized in the nucleus and overlapped with the mCherry signal (Figure 2A). The full length of TgMYB4 was cloned into the pGBKT7 vector and subsequently transferred into the yeast strain AH109 for transcriptional activation analysis. The yeast cell carrying the pGBKT7-TgMYB4 vector was able to grow in SD/-Trp/-Leu medium, but it was unable to grow in the selective medium SD/-Trp/-Leu/-Ade/-His with X-α-gal. This result demonstrated that *TgMYB4* did not display any transcriptional activation in yeast (Figure 2B).

We investigated the expression patterns of *TgMYB4* in different flower developmental stages of the yellow petal cultivar ‘Strong Golden’ and its budding cultivar ‘Strong Fire’ with red petals (Appendix A). The content of anthocyanin in ‘Strong Fire’ was significantly higher than that in ‘Strong Golden’. During petal coloration in ‘Strong Fire’, the expression level of *TgMYB4* was the lowest in S1 but showed significant increases in both S2 and S3, and ultimately peaked in S4. Conversely, in ‘Strong Golden’, *TgMYB4* exhibited a low expression in S1 and a slight increase in S2–S4. In summary, the expression level of *TgMYB4* increased following tulip petal coloration (S2–S4) and it was found to be higher in red petal cultivars as compared to yellow ones (Figure 2C).

### 2.2. TgMYB4 Negatively Regulates the Synthesis of Anthocyanin

The analysis of protein characterization and expression profiles implied that *TgMYB4* might play a negative role in anthocyanin synthesis. To confirm this observation, two independent *TgMYB4*-overexpressing transgenic tobacco lines (OE-1# and OE-2#) were generated. The petals of the control plants displayed a pink color, while the transgenic lines showed white or light pink petals (Figure 3A,B). The quantification assays revealed a significant decrease in total anthocyanin contents in the petals of *TgMYB4*-transgenic tobacco plants (Figure 3C). Furthermore, a qRT-PCR assay showed that genes involving in the anthocyanin synthesis pathway, including *NtCHS*, *NtCHI*, *NtF3H*, *NtF3′H*, *NtDFR*, and *NtANS*, were significantly down-regulated in *TgMYB4* transgenic lines (Figure 3D).

To further elucidate the role of *TgMYB4* in the synthesis of anthocyanin in tulips, a virus-induced gene silencing (VIGS) assay was carried out. The presence of TRV was confirmed through genomic PCR (Figure 4A). The qRT-PCR assay revealed that the expression level of *TgMYB4* was substantially suppressed in the silenced petals (Figure 4B). After 3 days of treatment, the petals of the TRV2 control started to display a red hue, whereas the *TgMYB4*-silenced petals displayed a more pronounced coloration. Additionally, the total anthocyanin content was significantly reduced in *TgMYB4*-silenced petals compared to the TRV2 control (Figure 4D). These results indicated that *TgMYB4* functioned as a negative regulator of anthocyanin biosynthesis.

### 2.3. TgMYB4 Suppresses the Activation of Anthocyanin Biosynthesis Genes Indirectly

To further characterize the working mechanisms of *TgMYB4*, we explored the regulatory relationship between *TgMYB4* and structural genes involved in anthocyanin biosynthesis. The promoters of *AtDFR* from *Arabidopsis thaliana* and *TfANS1* from *Tulipa fosteriana* were used as the targeted genes for further analyses [42,43].

For the dual-luciferase reporter assay, the full length of *TgMYB4* was combined into the pGreenII 62-SK as an effector, and the promoter fragments of *TfANS1* (-732-0 bp) and *AtDFR* (-520-0 bp) were cloned into the pGreenII 0800-LUC vector as reporters (Figure 5A). The luminescence live images revealed that the fluorescence signal was considerably weaker in the co-transformation of the *TgMYB4*-SK effector and the pTfANS1 or pAtDFR reporter compared to the control group. When *TgMYB4*-SK was co-transformed with pGreen0800-pTfANS1 and pGreen0800-pAtDFR, the ratio of LUC (firefly luciferase) to REN (Renilla luciferase) significantly decreased by 42.5% and 75.6%, respectively, compared to the control group. The observations of fluorescence signal and quantitative measurement of the LUC/REN ratio indicated that TgMYB4 was able to suppress the transcription activities of *TfANS1* and *AtDFR* (Figure 5B–E).

The Y1H assay was performed to investigate the interaction between TgMYB4 and promoters of *TfANS1* and *AtDFR*. The prey and bait vectors were constructed as shown in Figure 5F. The results showed that only the positive control grew normally on SD/-Trp/-Leu/-His media supplemented with 2.5 mM 3-AT. The yeast cells co-transformed with pGADT7-*TgMYB4* prey and the pHIS2-pAtDFR/pTfANS1 bait could not grow on the selected medium. The Y1H assay demonstrated that TgMYB4 could not bind to promoters of *TfANS1* and *AtDFR.* These results suggest that *TgMYB4* might inhibit the expression of structural genes through an indirect approach.

### 2.4. Protein Interaction between TgMYB4 and TgbHLH42-1

TgMYB4 contains a bHLH-interacting motif (Figure 1B), which was identified as a conserved domain of subgroup 4 R2R3-MYBs and was shown to interact with the bHLH component of the MBW complex. TgbHLH42-1 plays a positive role in the regulation of anthocyanin in tulips, and has a high homology to AtTT8 [44]. In order to dissect the working mechanism of *TgMYB4*, we explored the relationship between TgMYB4 and TgbHLH42-1.

Yeast two-hybrid assays were performed to investigate whether TgMYB4 could interact with TgbHLH42-1. The CDSs of *TgMYB4* and *TgbHLH42-1* were cloned and inserted into the pGBKT7 and pGADT7 vector as bait and prey plasmids, respectively (Figure 6A). All transformed yeast grew well on SD/-Trp/-Leu media, whereas only positive control and yeast cells harboring TgMYB4 bait and TgbHLH42-1 prey conducts survived and exhibited a blue color on SD/-Trp/-Leu/-Ade/-His media supplemented with X-α-gal (Figure 6B).

Moreover, a firefly luciferase complementation imaging assay (LCI) was conducted in tobacco leaves through transient overexpression of TgbHLH42-1 fused with the N-terminus of luciferase (nLUC) and TgMYB4 fused with the C-terminus of luciferase (cLUC) (Figure 6C). Through live luminescence imaging, significant LUC fluorescence signals were exclusively observed in the presence of TgbHLH42-1_nLUC and TgMYB4_cLUC, while marginal signals were detected in the other groups (Figure 6D). This result provided further evidence showing the interaction between TgMYB4 and TgbHLH42-1.

### 2.5. The Relationship between TgMYB4 and the MYB-bHLH-WD40 Complex

We previously demonstrated that the AtPAP1-TgbHLH42-1 complex could activate the transcription of *AtDFR* through dual luciferase assays [44]. In this study, we investigated whether TgMYB4 could attenuate the ability of the MYB-bHLH complex to activate *AtDFR*. The ratio of LUC to REN demonstrated that the expression of *AtDFR* was activated by AtPAP1, whereas it was repressed by AtMYB4 and unchanged in the presence of TgbHLH42-1. The highest ratio was observed when both AtPAP1 and TgbHLH42-1 were present. However, the addition of TgMYB4 led to a slight reduction in the LUC to REN ratio, with no clear dose-dependent effect observed. The evidence indicated that TgMYB4 could inhibit the activation effects of the MBW complex on the expression of biosynthesis-related genes (Figure 7A). Moreover, we investigated whether the function of TgMYB4 was influenced after interaction with TgbHLH42-1. As depicted in Figure 7B, with the increase in the TgbHLH42-1protein, the inhibitory impact of TgMYB4 on *AtDFR* diminished gradually and eventually ceased. The results presented above suggested that TgMYB4 might play an inhibitory role through competing with bHLH proteins in the MBW complex.

## 3. Discussion

Anthocyanins are a class of flavonoids that exhibit various functions such as attracting pollinators, defending against phytopathogens, protecting against UV light damage, and mitigating oxidative stress [5,45,46]. In ornamental plants, anthocyanins play a substantial role in red, purple, pink, and blue coloration, greatly contributing to their high commercial value [14,47,48]. Investigations into anthocyanin regulation began over 30 years ago with the cloning of the first transcription factor, *Colorless1* (*C1*), from maize [49,50]. Subsequently, investigations into other species, particularly *P. hybrida*, *Antirrhinum majus*, and *A. thaliana*, have significantly contributed to the advancement of the MBW model in understanding the regulation of anthocyanin [51,52,53,54,55]. In the past 20 years, the discovery of the anthocyanin repressor has improved our understanding of the regulatory mechanism of anthocyanin synthesis.

MYB repressors play important roles in regulating anthocyanin accumulation by providing feedback to the regulatory network [9,41,56]. In petunias, *PhMYB27* and *PhMYBX* serve to inhibit the production of anthocyanin, while their transcription was activated by the MBW complex. This illustrated a complex hierarchical and feedback regulation between the repressors and activating factors involved in anthocyanin regulation [9,38,57,58,59]. R2R3-MYB repressors, such as *MtMYB2* in *Medicago truncatula*; *PpMYB18* in *Prunus persica*; and *CsMYB3* in *Citrus sinensis*, typically exhibit high expressions during the anthocyanin-pigmented stage, and are activated by the MBW complex. It was suggested that the roles of these R2R3-MYB repressors might not be attributed to completely preventing anthocyanins synthesis, but to modulating the absolute anthocyanin intensity [9,19,20,29]. In this study, *TgMYB4* was characterized as an S4 R2R3-MYB anthocyanin repressor with a high expression level in anthocyanin-pigmented stages in tulip petals (Figure 1, Figure 2 and Figure 3). The LUC assay demonstrated that TgMYB4 could suppress the expression of anthocyanin-biosynthesis-related genes and attenuate the activating effects of AtPAP1 and TgbHLH42-1 (Figure 5). However, in the presence of both the TgMYB4 repressor and MYB activator, the activating factors play a prominent role, despite an abundance of TgMYB4 (Figure 7A). In summary, *TgMYB4* might suppress the transcription of anthocyanin-related genes to protect plants from excessive anthocyanin accumulation.

The R2R3 MYBs repressors are classified into two types according to the phylogenetic relationships: ‘*FaMYB1*-like’ and ‘*AtMYB4*-like’. In this study, *TgMYB4* was identified as an ‘*AtMYB4*-like’ repressor, which primarily regulates the general phenylpropanoid steps to reduce the accumulation of phenolic acids and lignans (Figure 1B). However, *TgMYB4* overexpression in tobacco plants reduced anthocyanin accumulation in flower petals, resulting in a white or light pink color. Moreover, the expression of the structural genes *NtCHS*, *NtCHI*, *NtF3H*, *NtF3′H*, *NtDFR*, and *NtANS* was consistently down-regulated in *TgMYB4* transgenic lines (Figure 3). In addition, the luciferase assay revealed the inhibition of TgMYB4 on the expression of *TfANS1* and *AtDFR* (Figure 5). Incidentally, narcissus (*Narcissus tazetta*) *NtMYB2* and apple (*Malus domestica*) *MdMYB16* were categorized in the ‘*AtMYB4*-like’ group; however, they inhibit the production of anthocyanin by suppressing the transcription of multiple genes in the anthocyanin pathway [13,60]. *PtMYB165* and *PtMYB194* in poplars fell into the ‘*FaMYB1*-like’ clade phylogenetically, and could greatly decrease the accumulation of various phenylpropanoids rather than flavonoids [61]. In general, the R2R3-MYB repressors typically play a key role in suppressing the biosynthesis of diverse metabolites in the phenylalanine pathway. Although these repressors exhibit a certain degree of division of labor, it is not strictly exclusive.

The highly conserved bHLH-binding motif and ERF motif are important for the suppression function of R2R3-MYB repressor proteins. For AtMYB4 and PpMYB18 (*P. persica*), mutations in either the bHLH-binding motif or the ERF motif significantly affected their inhibitory function. In poplars, PrtMYB182 without a functional bHLH-binding motif lost its repressor function, whereas mutations of the ERF motif did not lead to a noticeable decrease in repressor function [25]. On the contrary, the repressive function of MdMYB16 was unaffected by the removal of the bHLH-interacting motif. However, it was completely inactivated when the EAR motif was deleted [13]. Here, TgMYB4 contained a bHLH-binding motif, a C1 motif, and an ERF motif (Figure 1A). This protein demonstrated the capacity to interact with TgbHLH42-1 (Figure 6), and the inhibition of *AtDFR* expression was diminished in the presence of an excess amount of TgbHLH42-1 (Figure 7B). These results suggest that the physical interaction between TgMYB4 and TgbHLH42-1 plays a significant role in the repressive function of TgMYB4.

## 4. Materials and Methods

### 4.1. Plant Materials and Growth Conditions

The tulip cultivars ‘Strong Fire’ and ‘Strong Golden’ (Appendix A), *Nicotiana benthamiana*, and early flowering tobacco [62] were used in this study. The plant materials were cultivated in a plant growth room with a 16/8 h light/dark cycle and a temperature range of 18–22 °C.

### 4.2. Total RNA Extraction, Gene Cloning and Real-Time Quantitative PCR Analysis

Total RNA isolation, first-strand cDNA synthesis, gene cloning and qRT-PCR assays were performed using the same methods as described previously [63,64]. The specific primers were designed by the web tool Primer-BLAST on the NCBI (National Center for Biotechnology Information) website (https://www.ncbi.nlm.nih.gov/tools/primer-blast/index.cgi?LINK_LOC=BlastHome (accessed on 11 December 2023)). All the primer sequences are listed in Appendix A.

### 4.3. Multiple Sequence Alignment and Phylogenetic Analysis

Multiple protein sequence alignments were performed using the ClustalW1.81 and GenDoc 2.7 programs. The conserved domains were identified using the Conserved Domain Search Service (http://www.ncbi.nlm.nih.gov/Structure/cdd/wrpsb.cgi? (accessed on 11 December 2023)). The phylogenetic tree was generated using the maximum-likelihood method with 1000 bootstrap replicates in MEGA11.

### 4.4. Vector Construction and Plant Transformation

The coding sequence (CDS) of TgMYB4 was recombined into the pCAMBIA2300s vector under the control of the CaMV 35S promoter. *TgMYB4* transgenic tobacco lines were generated through leaf disc transformation [62]. All the primer sequences are listed in Appendix A.

### 4.5. VIGS Assay in Tulip Petals

The fragment located 313–624 bp downstream of the start codon (ATG) of TgMYB4 was amplified and cloned into the pTRV2 vector. The recombinant pTRV2-TgMYB4 vector, empty pTRV2, and pTRV1 vectors were transformed into A. tumefaciens GV3101. The petals of the red cultivar ‘SF’ were used for the assay before coloration. The specific steps of the assay were conducted following a previously described protocol [44,65]. Firstly, the petals were immersed in a bacterial suspension containing pTRV1 and pTRV2 or pTRV2-*TgMYB4* (OD600 = 1.5, Ratio = 1:1 *v*/*v*) and vacuum-infiltrated at 0.7 Mpa for 5 min. After infiltration, the petals were washed with distilled water and incubated in darkness at 22 °C for 24 h. All petals were then transferred to a growth chamber with a 16/8 h light/dark cycle for the specified time points. The primer sequences used in VIGS assay are listed in Appendix A.

### 4.6. Subcellular Localization

A subcellular localization assay was performed following the established protocol outlined in our previous work [44,65]. The ORF of *TgMYB4* without a stop codon was amplified and inserted into the L101YFP vector to generate the *35S::TgMYB4*-YFP construct. The construct was transformed into *A. tumefaciens* strain GV3101 for a transient assay in *N. benthamiana* leaves. After 2–3 days of culture, the leaves were observed thought a confocal laser scanning microscope (Leica TCSSP8). The primer sequences are listed in Appendix A.

### 4.7. Transcriptional Activation ACTIVITY Assay

For transcriptional activation analyses, the full length of TgMYB4 was amplified and cloned into the pGBKT7 vector containing the GAL4 DNA-binding domain. The recombinant plasmid, pGBKT7-p53, and empty vector pGBKT7 were transformed into yeast strain AH109. All the positive clones were plated onto SD/-Trp/-Leu and SD/-Trp/-Leu/-His/-Ade medium with or without X-α-gal and incubated at 30 °C for 48–72h. The primer sequences are listed in Appendix A.

### 4.8. Yeast One-Hybrid Assay

The promoter sequences of *TfANS1* and *AtDFR* were amplified and inserted into pHIS2 as the bait vectors, and the CDS of *TgMYB4* was cloned into pGADT7 as the prey vector. The bait and prey plasmids were transformed into the Y187 yeast strain and cultivated on SD/-Trp/-Leu selection plates at 30 °C. Then, the positive clones were transferred to SD/-Trp/-Leu and SD/-Leu/-Trp/-His/-Ade media with 2.5 mmol/L 3-AT and X-α-Gal. The primer sequences are listed in Appendix A.

### 4.9. Yeast Two-Hybrid Assay

For yeast two-hybrid assay, the CDSs of TgbHLH42-1 and TgMYB4 were cloned into pGADT7 and pGBKT7 vectors to generate the prey and bait plasmids. The recombinant plasmids were co-transformed into the AH109 strain and grown on SD/–Leu/–Trp medium at 30 °C for 3–4 days. Then, several dilutions of transformants were transferred to SD/-Leu/-Trp/-His/-Ade with 3-AT and X-α-Gal. The primer sequences are listed in Appendix A.

### 4.10. Dual Luciferase Assay

The CDSs of *TgMYB4*, *TgbHLH42-1,* and *AtPAP1* were amplified and ligated into the pGreenII 62-SK effector vector under the control of CaMV 35S. The *TfANS1* (GenBank: KC261507) and *AtDFR* promoter fragments were ligated into the pGreen II 0800-LUC vector as reporters. The vectors were introduced into *A. tumefaciens* strain GV3101 and transiently expressed in 4-week-old tobacco leaves as previously described by Hu et al. [44].

Luminescence live images were acquired using the plant living imaging system as described by Meng et al. [65]. The ratio of LUC to REN, determined using the Dual-Luciferase^®^ Reporter Assay System (Promega), was calculated to detect the transcriptional activity. The primer sequences are listed in Appendix A.

### 4.11. Firefly Luciferase Complementation Imaging Assay (LCI)

A firefly luciferase complementation imaging (LCI) assay was performed following the description of Chen et al. [66] and Li et al. [67]. The full lengths of TgbHLH42-1 and *TgMYB4* were inserted into pCAMBIA1300-nLUC and pCAMBIA1300-cLUC vectors to create TgbHLH42-1-nLUC and TgMYB4-cLUC constructs. The constructs or empty plasmids were transformed into *A. tumefaciens* GV3101 and transiently expressed in *N. benthamiana* leaves. The LUC fluorescence signal was observed thought a live plant imaging system. The primer sequences are listed in Appendix A.

### 4.12. Anthocyanin Extraction and Determination

The method of total anthocyanin extraction and determination was performed following to the description of Luo et al. [68] and Shen et al. [69]. The lyophilized powder of the petals (0.1 g) was extracted with 1 mL of extraction solution (0.1% HCl in methanol) at 4 °C in the dark for 24 h. After extraction, the samples were centrifuged at 12,000× *g* for 10 min, and the supernatant was collected for analysis. Finally, the absorbance values were determined via a multifunctional microporous plate detector (TECAN, Beijing, China) at wavelengths of 530 and 657. The content of anthocyanin was calculated as Q_Anthocyanins_ = (A_530_ − 0.25 × A_657_) × M^−1^, where A_530_ and A_657_ correspond to the absorption at specified wavelengths and M represents the weight of the plant material used for extraction.

### 4.13. Statistical Analysis

All experiments in this study were conducted a minimum of three times. The data were expressed as means ± standard error (SE). Statistical analysis was performed using a Student’s *t*-test, with significance levels of *p* < 0.05 (*) and *p* < 0.01 (**). Duncan’s test was employed to assess significant differences among multiple comparisons via PASS 26.0 software.

## 5. Conclusions

In conclusion, *TgMYB4* is an R2R3-MYB transcriptional repressor that acts to inhibit anthocyanin synthesis in tulip petals. The bHLH-binding motif in the R3 domain of TgMYB4 is crucial for its repression function, but further investigation is needed to understand its regulatory mechanisms.

## Figures and Tables

**Figure 1 ijms-25-00563-f001:**
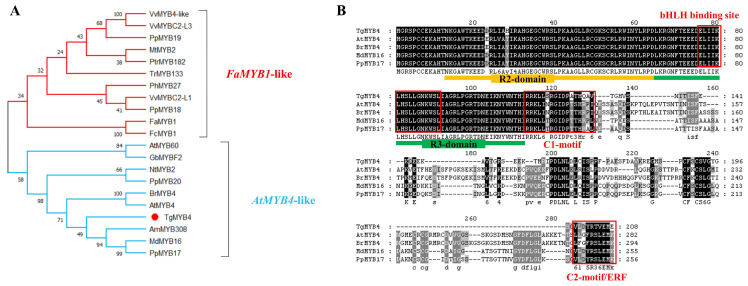
Analysis of the sequence structure and evolutionary characteristics of TgMYB4. (**A**) Phylogenetic analysis of relationships between TgMYB4 and R2R3-MYB subgroup 4 repressors in other plants. (**B**) Alignment of amino acid sequences of TgMYB4 compared with homologous proteins in other plant species. The orange color: R2-domain; the green color: R3-domain; the red frame: corresponding binding site or motif in the figure. The accession numbers for sequence data for phylogeny are VvMYB4-like (AID68565.1), VvMYBC2-L1 (NP_001268133), VvMYBC2-L3 (AIP98385) in *Vitis vinifera*; PpMYB17(ALO81020.1), PpMYB18 (ALO81021.1), PpMYB19 (ALO81022.1), PpMYB20(ALO81023.1) in *Prunus persica*; MtMYB2 (AES99346) in *Medicago truncatula;* PtrMYB182 (AJI76863) in *Populus tremula* × *Populus tremuloides*; TrMYB133 (AMB27081.1) in *Trifolium repens*; PhMYB27 (AHX24372) in *Petunia hybrida*; FaMYB1(AAK84064) in *Fragaria ananassa*; FcMYB1 (UXX19580.1) in *Fragaria chiloensis*; AtMYB4 (AEE86955.1), AtMYB60 (NP_001318958.1) in *Arabidopsis thaliana*; GbMYBF2 (AHI43788.1) in *Ginkgo biloba*; MdMYB16 (HM122617) in *Malus domestica*; NtMYB2 (ATO58377.1) in *Chinese narcissus*; BrMYB4 (ADZ98868.1) in *Brassica rapa*; AmMYB308 (P81393) in *Antirrhinum majus*.

**Figure 2 ijms-25-00563-f002:**
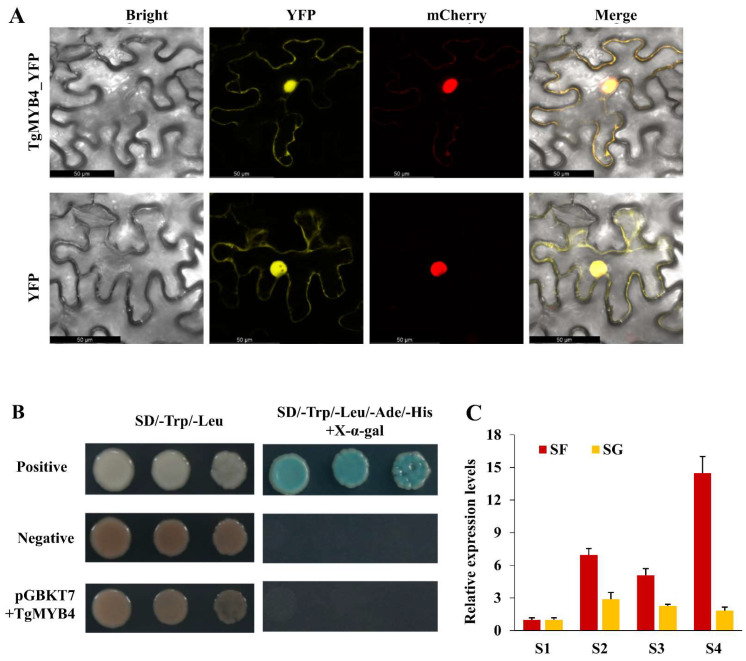
Characterization of *TgMYB4* during flower coloring in *T. gesneriana*. **(A)** Subcellular location of TgMYB4 in tobacco leaves. Confocal microscopic images show fluorescence signals of bright field, YFP (yellow), mCherry (red) in epidermal cells.. Merged: merged images of fluorescence signals in bright field. Scale bars = 50 μm. (**B**) Transcription activity assay. The positive and negative controls contain pGBKT7-p53 and pGBKT7 vectors, respectively. (**C**) The expression levels of *TgMYB4* during flower development in a red petal cultivar (SF: ‘Strong Fire’) and yellow petal cultivar (SG, ‘Strong Gol den’).

**Figure 3 ijms-25-00563-f003:**
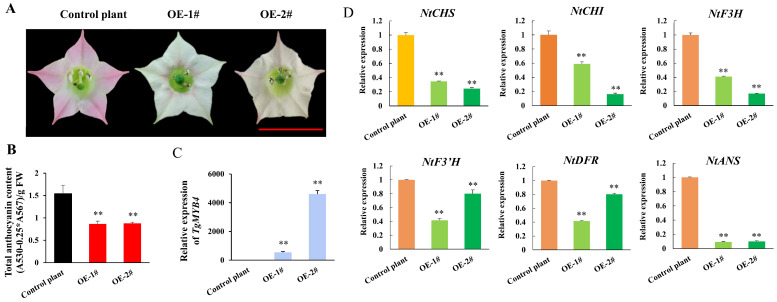
The phenotypes of *TgMYB4* transgenic tobaccos. **(A)** Pigmentation of flower petals of the control and *TgMYB4* transgenic tobacco plants. The scale bar=1cm. (B) Total anthocyanin content of the control and transgenic lines. (**C**) The expression level of *TgMYB4* in the control and transgenic tobaccos. (**D**) Expression levels of anthocyanin-related genes in the control and *TgMYB4* transgenic tobaccos. Error bars indicate SE (n = 3), ** *p* < 0.01.

**Figure 4 ijms-25-00563-f004:**
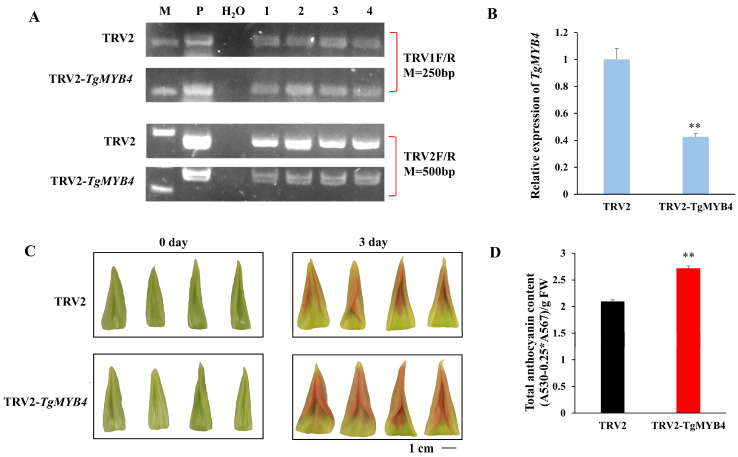
Simultaneously silencing TgMYB4 increased anthocyanin accumulation in tulip petals. **(A)** Genomic PCR of silenced petals. M: DNA marker, P: plasmid. (**B**) Expression levels of TgMYB4 in the petals of TRV2 control and *TgMYB4*-silenced petals. (**C**) The petal coloring phenotype of TRV2 control and *TgMYB4*-silenced petals. Scale bar = 1 cm. (**D**) The total anthocyanin contents of TRV2 control and *TgMYB4*-silenced petals. Error bars indicate SE (n = 4), ** *p* < 0.01.

**Figure 5 ijms-25-00563-f005:**
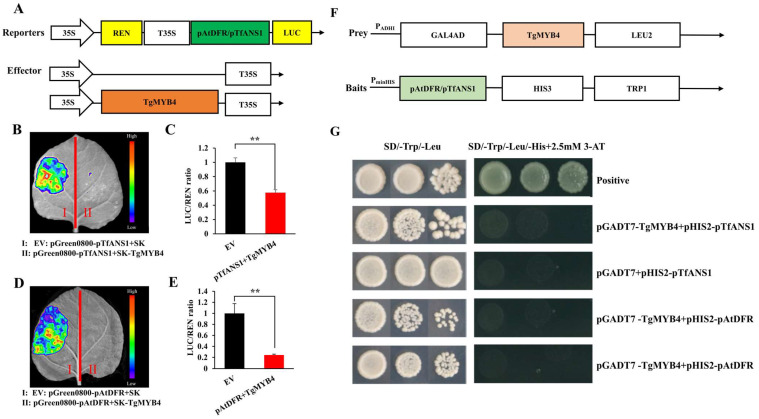
TgMYB4 indirectly represses the transcription of *TfANS1* and *AtDFR*. (**A**) A schematic of the dual luciferase (LUC) reporters and effectors. LUC, firefly luciferase; REN, Renilla luciferase. (**B**–**E**) Dual luciferase (LUC) assay showing TgMYB4 repressed the activation of *TfANS1* and *AtDFR* in *N. benthamiana* leaves. SK empty vector + pGreen0800- pTfANS1/pAtDFR was used as the negative control. (**F**) A schematic of prey and bait used for yeast one-hybrid assays. (**G**) TgMYB4 could not bind to the promoter of *TfANS1* or *AtDFR* thought yeast one-hybrid assays. 3AT: 3-amino-1,2,4-triazole. pTfANS1: the promoter of *ANS1* in *Tulipa fosteriana*; pAtDFR the promoter of *DFR* in *Arabidopsis thaliana.* Error bars indicate SE (n = 3), ** *p* < 0.01.

**Figure 6 ijms-25-00563-f006:**
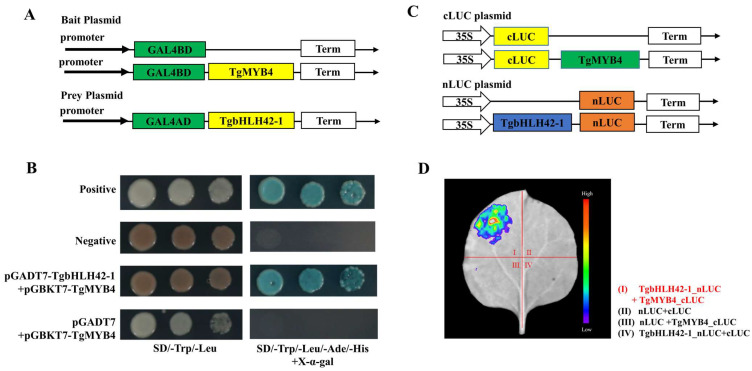
Protein interactions between TgMYB4 and TgbHLH42-1. (**A**) Schematic diagram for bait and prey in a yeast two-hybrid assay. (**B**) TgMYB4 interacts with TgbHLH42-1 in yeast two-hybrid assays. The positive and negative control contain pGADT7-T with pGBKT7-p53 or pGBKT7-lam vectors, respectively. (**C**) Schematic diagram for c-LUC and n-LUC plasmids. (**D**) Bimolecular luminescence complementation assay demonstrating that TgMYB4 interacts with TgbHLH42-1 in *N. benthamiana* leaves.

**Figure 7 ijms-25-00563-f007:**
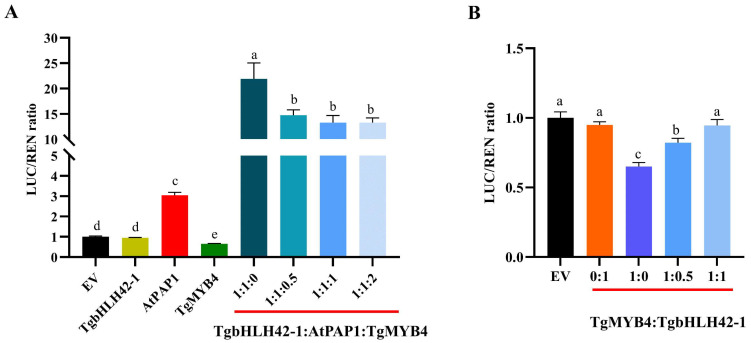
Dual luciferase assay to test the activation and repression of the *AtDFR* promoter by repressor MYB, activator MYB and bHLH factors. (**A**) TgMYB4 inhibited the activation of the MYB-bHLH complex on the *AtDFR* promoter. (**B**) Activation effect of co-infiltration of TgMYB4 and TgbHLH42-1 at different ratios on the *AtDFR* promoter. LUC, firefly luciferase; REN, Renilla luciferase. SK empty vector + pGreen0800. pAtDFR was used as the negative control. Error bars indicate SE (n = 3). Different letters indicate significant differences analyzed using Duncan’s multiple comparison tests.

## Data Availability

All data supporting the findings of this study are available within the paper and its Appendix A published online.

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
