# Peer review of "The R2R3-MYB Transcriptional Repressor TgMYB4 Negatively Regulates Anthocyanin Biosynthesis in Tulips (Tulipa gesneriana L.)"

_ijms, 2024, doi:10.3390/ijms25010563_

Round 1

Reviewer 1 Report

Comments and Suggestions for Authors

 This article is entitled “A R2R3-MYB transcriptional repressor TgMYB4 negatively regulates anthocyanin biosynthesis in tulip”. Flower color is an important horticulture trait for ornamental crops. It is an interesting finding and the authors provided detailed molecular data. However, some inconsistent results in this paper.

Comments :

1.          The authors showed TgMYB4 transcript is highly expressed in red color tulip cultivar compared to the yellow cultivar (Fig.2C). It seems TgMYB4 is a positive regulator of red flower. Red tulip has higher TgMYB4 than yellow cultivar. However, on Page 4 authors said “2.2 TgMYB4 negatively regulates the synthesis of anthocyanin”. Their conclusion is based on the results of heterologous overexpressing tulip’s TgMYB4 in transgenic tobacco. Upregulation of TgMYB4 has reduced red/ pink color in transgenic tobacco (Fig.3). I wonder whether this is due to the overexpression of TgMYB4 in a heterologous system of tobacco leading to a competition effect. Or does tobacco lack a specific regulator ??

2.          That would be nice if the authors could transiently overexpess TgMYB4 in a yellow tulip flower or knock down in a red tulip… So far, this article has not provided solid data to show “TgMYB4 negatively regulates (?) anthocyanin biosynthesis in tulip”. I doubt the title of this article is solid or conclusive.

3.          For the Conclusions section, “In conclusion, TgMYB4 is a R2R3-MYB transcriptional repressor that acts to inhibit anthocyanin synthesis in tulip petals.” Need to be aware of this conclusion.  

4.          I suggest authors change the order of Fig.3B with 3C. B is the anthocyanin content. C is the RT-qPCR data of target gene TgMYB4. Then, shows the upregulation TgMYB4 transcript in tobacco altered the anthocyanin biosynthesis genes of tobacco afterwards.   

Author Response

Response to Reviewer 1 Points

Dear Reviewer,

We appreciate it for your valuable comments and suggestions. We thoroughly checked the manuscript and carefully answered all of questions raised. For detail, please see the following answers. If you have any queries, please don’t hesitate to contact me.

Best regards!

Lin Xiang

[email protected]

 Comment 1 The authors showed TgMYB4 transcript is highly expressed in red color tulip cultivar compared to the yellow cultivar (Fig.2C). It seems TgMYB4 is a positive regulator of red flower. Red tulip has higher TgMYB4 than yellow cultivar. However, on Page 4 authors said “2.2 TgMYB4 negatively regulates the synthesis of anthocyanin”. Their conclusion is based on the results of heterologous overexpressing tulip’s TgMYB4 in transgenic tobacco. Upregulation of TgMYB4 has reduced red/ pink color in transgenic tobacco (Fig.3). I wonder whether this is due to the overexpression of TgMYB4 in a heterologous system of tobacco leading to a competition effect. Or does tobacco lack a specific regulator??

Answer: Thank you for comments. In this study, TgMYB4 shows higher expression levels in in red color tulip cultivar, which aligns with the findings of many R2R3-MYB repressors reported in other species. And the further elaboration on this point can be found in the second paragraph of the discussion section at L285-L290. “The R2R3-MYB repressors, such as MtMYB2 in Medicago truncatula; PpMYB18 in Prunus persica; CsMYB3 in Citrus sinensis, typically exhibited high expression during the anthocyanin-pigmented stage, and were activated by the MBW complex. It was suggested that the roles of these R2R3-MYB repressors might not be attributed to prevent anthocyanins synthesis thoroughly, but to modulate the absolute anthocyanin intensity [9,19,20,29].”

The protein structure of TgMYB4 exhibits distinct repressor motif shown in Figure 1. Moreover, it was observed that TgMYB4 effectively represses the expression of TfANS1 in Tulipa fosteriana, through the LUC assay (Figure 4). We suggest that the light pink or white petals observed in TgMYB4-transgenic tobaccos are probably due to the suppression of gene expression involved in the anthocyanin synthesis pathway by TgMYB4.

 Comment 2 That would be nice if the authors could transiently overexpess TgMYB4 in a yellow tulip flower or knock down in a red tulip… So far, this article has not provided solid data to show “TgMYB4 negatively regulates (?) anthocyanin biosynthesis in tulip”. I doubt the title of this article is solid or;conclusive.

Answer: Thank you for comments. We have performed the VIGS assay to repress the expression of TgMYB4 in petals of red tulip. After treatment, the TgMYB4-silenced petals displayed a more pronounced coloration compared to the control. The result was shown in Figure 4 and described at L162-L170 in the text.

 Comment 3: For the Conclusions section, “In conclusion, TgMYB4 is a R2R3-MYB transcriptional repressor that acts to inhibit anthocyanin synthesis in tulip petals.” Need to be aware of this conclusion.  

Answer: Thank you for comments. After carefully analyzing several factors related to TgMYB4, including its protein structure repressor motifs, expression patterns, transgenic tobacco phenotypes, results of VIGS assay, and inhibitory effects on downstream genes, we maintain the belief that TgMYB4 negatively regulates the synthesis of anthocyanin in tulip petals.

 Comment 4: I suggest authors change the order of Fig.3B with 3C. B is the anthocyanin content. C is the RT-qPCR data of target gene TgMYB4. Then, shows the upregulation TgMYB4 transcript in tobacco altered the anthocyanin biosynthesis genes of tobacco afterwards.   

Answer: Thank you for suggests. We have changed the order of Fig.3B with 3C.

Reviewer 2 Report

Comments and Suggestions for Authors

The manuscript explores the mechanisms behind anthocyanin regulation in plants, focusing on the identification and characterization of a specific transcription factor as a repressor of anthocyanin synthesis in tulip petals and its impact on anthocyanin synthesis or related pathways. The study highlights the complex regulatory network governing anthocyanin synthesis and provides insights into how specific transcription factors like TgMYB4 modulate this process.

For an improved version, please consider the following suggestions:

P.1 – provide the explanation before the first use of MYB acronym

P.2, figure 2 – please consider shorten the figure caption and incorporate the discussed points into the main text instead

P.5 – provide the explanation before the first use of REN acronym

#3 – an important part of text in this section has to be re-located to introduction; retain only information directly associated with our findings concerning previous studies

#5.11 – provide relevant experimental details about sampling, sample preparation, sample amounts, reagents (types and providers|), instruments (types and providers).

Comments on the Quality of English Language

Only minor editing of English language & punctuation are required

Author Response

Response to Reviewer 2 Points

Dear Reviewer,

We appreciate it for your valuable comments and suggestions. We thoroughly checked the manuscript and carefully answered all of questions raised. For detail, please see the following answers. If you have any queries, please don’t hesitate to contact me.

Best regards!

Lin Xiang

[email protected]

Comment: P.1 – provide the explanation before the first use of MYB acronym

Answer: Thank you for suggestion. We have explained the MYB acronym when it first used in the introduction section at L35.

Comment: P.2, figure 2 – please consider shorten the figure caption and incorporate the discussed points into the main text instead

Answer: Thank you for suggestion. We have shorted the caption of Figure 2 and added some points into the main text at L128-L135.

Comment: P.5 – provide the explanation before the first use of REN acronym

Answer: Thank you for suggestion. We have explained the REN acronym at L195.

Comment: #3 – an important part of text in this section has to be re-located to introduction; retain only information directly associated with our findings concerning previous studies

Answer: Thank you for suggestion. We have relocated some portions of the text originally found in the third and fourth paragraphs of the discussion section to L57-L63 and L70-L78 of the introduction.

#5.11 – provide relevant experimental details about sampling, sample preparation, sample amounts, reagents (types and providers|), instruments (types and providers).

Answer: Thank you for suggestion. We have supplemented the pertinent information about this experiment at L410-L419.

Comments on the Quality of English Language

Only minor editing of English language & punctuation are required.

Answer: We thoroughly checked the manuscript to improve the English language and punctuation.

Reviewer 3 Report

Comments and Suggestions for Authors

Dear Authors!

The regulation of anthocyanin biosynthesis gene expression has been fairly well studied to date. The role of the conserved MYB-bHLH-WD40 (MBW) complex involved in the regulation of transcription has been studied in detail, negative regulators of gene transcription of the anthocyanin biosynthesis pathway have been isolated and characterized, but the mechanism of their negative regulation is still unclear. In such a situation, the manuscript under review is quite timely. The authors isolated, characterized structurally and largely functionally a novel repressor of anthocyanin biosynthesis from Tulipa gesneriana, which was designated as TgMYB4. The gene for this repressor had the highest expression level in the S4 stage with maximum anthocyanin accumulation in the 'Strong Fire' cultivar. In transgenic tobaccos in which the TgMYB gene was overexpressed, the anthocyanin content was significantly reduced. In addition, TgMYB4 suppressed the expression of a number of anthocyanin biosynthesis genes. But, very interestingly, according to the authors' data TgMYB4 suppressed the expression of anthocyanin biosynthesis genes not directly, that is, without interacting with their promoters. A large number of very modern analytical methods were used to study the interaction of TgMYB4 with other transcription regulator proteins as well. Based on these results, the authors hypothesize (have some experimental confirmation) that the TgMYB4 protein may compete with the MYB activator for binding to the major Helix-Loop-Helix (bHLH) protein, resulting in suppression of the function of the MBW complex.

Judging from the interaction between TgMYB4 and TgbHLH42-1, the TgMYB4 repressor does not result in complete suppression of TgbHLH42-1 activity. However, when transgenic tobacco plants in which the TgMYB4 gene was overexpressed, a very strong suppression of anthocyanins content was observed. This may mean that the repressor prevents maximal anthocyanin accumulation under some conditions and suppresses anthocyanin accumulation almost completely under other conditions.

The article is very good. From my understanding of those presented, the article can be accepted for publication almost as is and hopefully it will be an ornament to Plant Molecular Biology part in the Journal..

I have no remarks to make. I think the sentence “When TgMYB4-SK was co-transformed with pGreen0800-pAtDFR and pGreen0800-pTfANS1, the ratio of LUC to REN significantly decreased by % and 42.4%, respectively, compared to the control group" in part 2.3 is unfortunate”.

Best wishes

Victor

Author Response

Response to Reviewer 3 Points

Dear Reviewer,

Thank you very much for your recognition of our work. We have thoroughly checked the manuscript to improve the manuscript. Additionally, we apologize for our mistake, and it has already been corrected. ‘When TgMYB4-SK was co-transformed with pGreen0800-pTfANS1 and pGreen0800-pAtDFR, the ratio of LUC (firefly luciferase) to REN (Renilla luciferase) significantly decreased by 42.5% and 75.6%, respectively, compared to the control group.’ If you have any queries, please don’t hesitate to contact me.

Best regards!

Lin Xiang

[email protected]

Round 2

Reviewer 1 Report

Comments and Suggestions for Authors

Thanks for making the revisions and the manuscript is improved.

I have no other comments.